# Phospho-Specific Flow Cytometry Reveals Signaling Heterogeneity in T-Cell Acute Lymphoblastic Leukemia Cell Lines

**DOI:** 10.3390/cells11132072

**Published:** 2022-06-29

**Authors:** Omar Perbellini, Chiara Cavallini, Roberto Chignola, Marilisa Galasso, Maria T. Scupoli

**Affiliations:** 1Department of Cell Therapy and Hematology, San Bortolo Hospital, Viale Ferdinando Rodolfi, 37, 36100 Vicenza, Italy; omar.perbellini@aulss8.veneto.it; 2Research Center LURM, Interdepartmental Laboratory of Medical Research, University of Verona, Piazzale L.A. Scuro, 10, 37134 Verona, Italy; chiara.cavallini@univr.it; 3Department of Biotechnology, University of Verona, Strada Le Grazie 15, 37134 Verona, Italy; roberto.chignola@univr.it; 4Department of Neurosciences, Biomedicine and Movement Sciences, University of Verona, Piazzale L.A. Scuro, 10, 37134 Verona, Italy; marilisa.galasso@univr.it

**Keywords:** cell signaling, leukemia, flow cytometry

## Abstract

Several signaling pathways are aberrantly activated in T-ALL due to genetic alterations of their components and in response to external microenvironmental cues. To functionally characterize elements of the signaling network in T-ALL, here we analyzed ten signaling proteins that are frequently altered in T-ALL -namely Akt, Erk1/2, JNK, Lck, NF-κB p65, p38, STAT3, STAT5, ZAP70, Rb- in Jurkat, CEM and MOLT4 cell lines, using phospho-specific flow cytometry. Phosphorylation statuses of signaling proteins were measured in the basal condition or under modulation with H_2_O_2_, PMA, CXCL12 or IL7. Signaling profiles are characterized by a high variability across the analyzed T-ALL cell lines. Hierarchical clustering analysis documents that higher intrinsic phosphorylation of Erk1/2, Lck, ZAP70, and Akt, together with ZAP70 phosphorylation induced by H_2_O_2_, identifies Jurkat cells. In contrast, CEM are characterized by higher intrinsic phosphorylation of JNK and Rb and higher responsiveness of Akt to external stimuli. MOLT4 cells are characterized by higher basal STAT3 phosphorylation. These data document that phospho-specific flow cytometry reveals a high variability in intrinsic as well as modulated signaling networks across different T-ALL cell lines. Characterizing signaling network profiles across individual leukemia could provide the basis to identify molecular targets for personalized T-ALL therapy.

## 1. Introduction

T-cell acute lymphoblastic leukemia (T-ALL) is a malignancy of immature T cells at defined stages of intrathymic T-cell differentiation [1]. T-ALL accounts for about 15% of pediatric and 25% of adult ALL cases [2,3]. Over the past last decades, survival rates of patients with T-ALL have remarkably improved thanks to advances in combined chemotherapy protocols [4]. However, primary resistance to treatment and relapse are observed in a significant number of patients [5]. The poor outcome of T-ALL patients has been related to disease complexity, where genetic and microenvironmental factors interact in supporting leukemia progression and resistance [6]. Therefore, for a comprehensive classification of the disease and for developing personalized therapeutic strategies, it is challenging to gain an integrated, overall picture of changes of the entire leukemia cell signaling.

Immunophenotyping allows the classification of T-ALL in different subgroups based on the corresponding stage of T-cell ontogenesis [7]. The prognostic value of different immunophenotypic groups differs between the studies, also in the case of early T-cell precursor ALL (ETP-ALL), which is a very unique subgroup in T-ALL [8,9,10]. At the molecular level, T-ALL is a heterogenous disease characterized by a wide spectrum of genetic lesions [11] as well as by altered interpretation of external cues deriving from the microenvironment [6]. Both cell-intrinsic defects and microenvironmental stimuli converge on the activation of regulatory signaling pathways involved in processes enhancing the capacity of self-renewal, overturning the control of cell proliferation, blocking differentiation, and promoting resistance to apoptosis. Signaling pathways that are aberrantly activated in T-ALL include phosphatidylinositol-3OH-kinase (PI3K)/Akt, mitogen-activated protein kinases (MAPKs), and Janus kinases/signal transducer activator of transcription Jak/STAT [12,13]. These signaling pathways can be activated due to genetic alterations of their components as well as in response to external microenvironmental cues. Amongst cell-extrinsic modulators, the chemokine CXCL12 and interleukin 7 (IL7) play a key role in promoting tumor growth [14,15,16,17,18].

CXCL12 is a homeostatic chemokine secreted by stromal cells, fibroblasts and epithelial cells within tissue microenvironments, including lymph nodes and bone marrow (BM) [19,20,21]. CXCL12 binds the CXC-chemokine receptors 4 (CXCR4) and 7 (CXCR7) [22]. Interaction of CXCL12 with CXCR4 triggers complex signaling pathways, which include PI3K/Akt, the MAPK extracellular signal-regulated kinase (Erk)1/2 and p38, and Jak/STAT, regulating intracellular calcium flux, chemotaxis, transcription and cell survival [23,24,25]. Evidence indicates that the CXCL12/CXCR4 signaling axis plays a role in cancer biology, in solid tumors as well as in hematological malignancies such as T-ALL, where the CXCL12/CXCR4 signaling regulates the maintenance and progression of leukemia [18,26,27,28,29]. The CXCL12/CXCR7 binding induces intracellular signaling pathways, such as PI3K/Akt, MAPK, Jak/STAT3 [22]. However, the role of CXCR7 in cancer progression is controversial, as some reports suggest pro-metastatic responses, while others indicate inhibition of metastasis [19]. In T-ALL, CXCR7 has been suggested to contribute to leukemic-cell migration, potentiating the CXCL12/CXCR4 axis [30].

IL7 is a cytokine produced by stromal cells in the BM, thymus and other tissues. The IL7 receptor (IL7R) is expressed mainly by lymphoid cells [31]. The IL7/IL7R interaction is required for normal T-cell development and homeostasis of mature T-cells [31]. IL7/IL7R-mediated binding activates three main signaling pathways: Jak/STAT, MAPK/Erk and PI3K/Akt/mammalian target of rapamycin (mTOR) [32] in both T-ALL and healthy T-cells [33,34]. However, while PI3K/Akt/mTOR signaling mediated by IL7 promotes proliferation and survival of T-ALL cells by inducing downregulation of p27kip1 and upregulation of Bcl-2 [35,36], in healthy T cells PI3K signaling activation by IL7 induces cell cycle progression without influencing cell survival [37,38].

To gain overall picture of intrinsic and extrinsic signaling in T-ALL cells, in this study we investigated signaling pathways in three T-ALL cell lines, namely CEM, MOLT4 and Jurkat, using phospho-specific flow cytometry that simultaneously determines protein expression and protein post-translational modifications (i.e., phosphorylation) at a single cell level [39,40,41,42]. Our results show that single-cell signaling profiles measured by phospho-specific flow cytometry allow us to reveal heterogeneous signaling profiles in the analyzed T-ALL cell lines.

## 2. Materials and Methods

### 2.1. Cell Samples

The cryopreserved T-ALL cell lines CEM, MOLT4 and Jurkat (American Type Culture Collection, ATCC) were cultured in medium RPMI-1640 (GIBCO; Thermo Fisher Scientific, Waltham, MA) with 10% fetal bovine serum (FBS, GIBCO) at 37 °C in a humidified atmosphere containing 5% CO_2_. Peripheral mononuclear cells (PBMC) were obtained from heparin-anticoagulated leukocyte concentrate samples collected as from healthy donors (HDs) at the Transfusion Medicine Unit, Azienda Ospedaliera Universitaria Integrata (AOUI), Verona, Italy. In accordance with the Declaration of Helsinki, all blood donors provided written informed consent for the anonymous collection and use of their leukocyte concentrate samples—as byproduct of the manufacturing of *blood donation*- for research purposes. 

PBMCs were isolated by Ficoll-hypaque centrifugation (Lymphoprep; Nicomed, Oslo, Norway) and cryopreserved in liquid nitrogen. Thawing of T-ALL cell lines and PBMC from HDs was performed at 37 °C and transferred in a 1:1 solution of RPMI-1640 complete medium and heat inactivated FBS. Cells were pelleted at 400 g for 5 min and resuspended in prewarmed (37 °C) complete RPMI-1640 medium. Before use, viability of the cells was over 90%, as assessed by flow cytometry analysis after staining with 7-amino-actinomycin D (7-AAD; BD Biosciences, BD Biosciences, San Jose, CA, USA).

### 2.2. Immunophenotype Characterization

Immunophenotype characterization of CEM, MOLT4, and Jurkat cell lines was performed as previously described [43]. T-ALL cell lines were classified according to their maturation stage using the criteria defined by the European Group for Immunological Characterization of Leukemias (EGIL) [7]. Cell surface expression of CD127 and CD184 was detected by staining cells with fluorochrome-conjugated antibodies in the dark at room temperature for 15 min. The complete list of antibodies used is reported in Appendix A. Autofluorescence controls were used to set background signals. Samples were then washed and analyzed on a FACSCanto cytometer (Becton Dickinson, Franklin Lakes, NJ, USA). Approximately 10,000 gated events were acquired for each sample.

### 2.3. Cell Treatments

Before treatments, cell lines and PBMC from HDs were stained with LIVE/DEAD Fixable Near-IR stain (Invitrogen; Thermo Fisher Scientific, Waltham, MA, USA) according to the manufacturer’s instructions. Briefly, cells were collected, washed, counted, resuspended in PBS at 1 × 10^6^/mL and stained with the reactive dye in the dark at room temperature for 30 min. After two washes with PBS, cell lines and PBMC were resuspended in RPMI-1640 with 10% fetal bovine serum at 2 × 10^6^/mL and 4 × 10^6^/mL, respectively; divided in 5 tubes and rested at 37 °C in a humidified atmosphere containing 5% CO_2_ for 1 h. Each tube was then treated with a different modulator for 15 min (Appendix A) for phospho-specific flow cytometry. Unmodulated cells were used as control.

### 2.4. Phospho-Specific Flow Cytometry

Phospho-specific flow cytometry was performed as previously described [40]. Briefly, after 15 min of treatment, cells were fixed with pre-warmed Fix Buffer I (Becton Dickinson) at 37 °C for 10 min and permeabilized with ice-cold 50% methanol for 30 min on ice. Permeabilized cells were washed with PBS containing 0.1% bovine serum albumin and stained with fluorochrome-conjugated antibodies in the dark at room temperature for 40 min. The complete list of antibodies used, and the staining panels are reported in Appendix A, respectively. Isotype controls were used to set background signals. Three independent experiments were performed for each cell line and for HD T cells. Approximately 10,000 gated events were acquired for each sample on a FACSCanto cytometer (Becton Dickinson).

### 2.5. Flow Cytometry Data Analysis

Flow cytometry data were processed, analyzed, and graphed using FlowJo software (v10.8; TreeStar, Ashland, OR, USA). Dead cells, debris and doublets were excluded based on forward-scatter, side-scatter, and LIVE/DEAD signals. All analyses were performed on gated T cells, identified as CD45/CD3 co-expressing cells in HD samples and as cCD3 or CD7 positive in T-ALL cell lines. Median fluorescence intensity (MFI) was normalized with respect to MFI of isotype control (RMFI) and, when indicated, transformed into logBasal phosphorylation statuses of signaling proteins were considered activated when RMFI >1. The response of cells to treatments was calculated as log_2_Fold change [log_2_(RMFI modulated/RMFI unmodulated)], as previously described [40]. A log_2_ value equal to 0 corresponds to ‘no change’ with respect to the unstimulated. The response to external modulators was considered positive when log_2_Fold change > 0. We used the term “node” to refer to a proteomic readout in the presence or absence of a specific modulator. For example, the response to CXCL12 modulation can be measured using pErk1/2 as a readout. That node is designated “CXCL12-pErk1/2” [40].

### 2.6. Statistical Analysis

The unsupervised hierarchical cluster analysis (HCA) was performed using webtool ClustVis (v. 2.0, https://biit.cs.ut.ee/clustvis/; accessed on 21 September 2021) [44], calculating the Euclidean distances as the distance metric and the complete linkage as linkage method of analysis. Comparison of phosphorylation levels among T-ALL cell lines and healthy donor (HD) T cells were performed using the Kruskal–Wallis test and each *p* value was corrected for multiple comparisons using the Dunn’s test. Differences were considered statistically significant when *p* ≤ 0. Graphing and statistical analyses were performed using GraphPad Prism software (v. 7.03, GraphPad Software Inc., San Diego, CA, USA; accessed on 16 December 2021).

## 3. Results

### 3.1. Study Design

The phosphorylation levels of ten signaling proteins that are frequently altered in T-ALL (namely Akt, MAPK Erk1/2, c-Jun NH2-terminal protein kinase (JNK), lymphocyte-specific protein tyrosine kinase (Lck), nuclear factor kappa-light-chain-enhancer of activated B cells (NF-κB) p65, p38, STAT3, STAT5, zeta-chain-associated protein kinase 70 (ZAP70), and retinoblastoma protein (Rb)) were analyzed at the single-cell level in the T-ALL cell lines CEM, MOLT4, and Jurkat, using phospho-specific flow cytometry (Figure 1). Based on EGIL classification criteria [7], the T-ALL cell-lines were classified into categories corresponding to three differentiation stages, i.e., CEM corresponds to the T-II (pre-T) stage, MOLT4 to the T-III (cortical) stage, and Jurkat to the T-IV (mature) stage (Table 1). Phosphorylation statuses of signaling proteins were measured in the basal condition (i.e., unmodulated) or under modulation with H_2_O_2_, phorbol 12-myristate 13-acetate (PMA), CXCL12, or IL7.

### 3.2. Phosphorylation Profiles in the Basal, H_2_O_2_-, and PMA-Modulated Conditions

The basal phosphorylation statuses of signaling proteins were highly heterogenous among cell lines (σ^2^ across the phosphoproteins was 2.7 for CEM, 6.0 for MOLT4, 20.7 for Jurkat; Figure 2A,B). All T-ALL cell lines showed basal phosphorylation (RMFI > 1.5) of Erk1/2, Lck, NF-κB p65, STAT3, and ZAP70^Y319^, with higher levels of Erk1/2, Lck, and ZAP70^Y319^ in Jurkat cells. JNK was phosphorylated in CEM and Jurkat cells. Rb was constitutively phosphorylated in CEM and MOLT4. Basal phosphorylation of Akt was detected in CEM and Jurkat cells, with higher levels in Jurkat. ZAP70^Y292^ was constitutively phosphorylated in Jurkat cells. Pp38 and pSTAT5 levels were undetectable in all the analyzed T-ALL cell populations (Figure 2A,B). Overall, the levels of phosphorylation—expressed as the sum of phosphoprotein RMFI—were higher in Jurkat than in CEM and MOLT4 (the sum of RMFI was 33.12 for CEM; 30.54 for MOLT4; and 56.88 for Jurkat).

Then, we investigated the responsiveness of T-ALL cell lines to H_2_O_2_, which reversibly inhibits tyrosine phosphatases thereby shifting the equilibrium of phosphorylation reaction in favor of kinase activation [45]. Responses to H_2_O_2_ were measured as the log_2_Fold change in phosphorylation between H_2_O_2_-modulated and unmodulated conditions. As shown in Figure 2C,D, H_2_O_2_ perturbed signaling profiles in all the analyzed leukemic cell populations. Specifically, H_2_O_2_ induced increased phosphorylation of STAT3 in all T-ALL cell lines, with a higher response of CEM compared with MOLT4 and Jurkat. Moreover, H_2_O_2_ induced a clear-cut increase of pAkt in CEM and MOLT4; of pErk1/2 in CEM and Jurkat cells; of NF-κB p65 in MOLT4 cells; of pZAP70^Y319^ and pZAP70^Y292^ in Jurkat cells. Overall, the phosphorylation responses to H_2_O_2_—expressed as the sum of phosphoprotein responses—were higher in Jurkat cells and lower in CEM and MOLT4 (the sum of log_2_Fold change in phosphorylation between H_2_O_2_-modulated and unmodulated conditions was 1.86 for CEM; 3.98 for MOLT4; 5.89 for Jurkat).

Next, we stimulated T-ALL cell lines with PMA, which activates T cells through protein kinase C (PKC) [46]. As shown in Figure 2E,F, PMA induced a clear-cut increase of pSTAT3 in all T-ALL cell lines, with higher phosphorylation levels in CEM cells. Moreover, PMA induced augmented phosphorylation of Erk1/2 in CEM and Jurkat, and JNK in MOLT4.

Taken together, these data reveal high signaling variability across T-ALL cell lines with distinct phosphorylation profiles for each cell population and specific signaling sensitivity to phosphatase inhibition and to PKC activation.

### 3.3. Phosphorylation Profiles Following Modulation with CXCL12 or IL7

We examined the responses of phosphoproteins to some physiological stimuli relevant for T-ALL pathophysiology, i.e., CXCL12 and IL7, in T-ALL cell lines. All T-cell populations expressed the CXCL12 receptor CXCR4 (CD184), with T-ALL cell lines showing higher levels than HD T cells, used as control (Appendix A). Moreover, all T-ALL populations expressed very low levels of IL7Rα chain (CD127) (Appendix A). Responsiveness of T-ALL cell lines to external stimuli was measured as the log_2_Fold change in phosphorylation between modulated and unmodulated conditions (Figure 3). CXCL12 evoked a clear-cut increase of pAkt in CEM and MOLT4 and of pJNK in MOLT4 (Figure 3A,B). Engagement of IL7R with IL7 induced augmentation of pAkt in MOLT4 cells (Figure 3C,D). In contrast, STAT5 was unresponsive to IL7 in T-ALL cell lines, but not in HD T cells used as control (Figure 3C,D).

### 3.4. Signaling Profiles Separate T-ALL Cell Lines in Distinct Clusters

Basal phosphorylation statuses of signaling proteins—measured as log_2_ RMFI—and responses to external stimuli—measured by the log_2_Fold change in phosphorylation between modulated and unmodulated conditions—were subjected to unsupervised hierarchical clustering analysis (HCA) within the three T-ALL cell populations. Unsupervised HCA of all the analyzed nodes (*n* = 55) identified CEM/MOLT4 cells and Jurkat as separate clusters (Appendix A). To reduce the number of nodes that can discriminate T-ALL cell populations, next we performed an unsupervised HCA on selected nodes showing the greatest variation across samples (σ^2^ of log_2_ values > 0.5, *n* = 12). Remarkably, this clustering preserved the classification obtained analyzing all nodes and allowed us to identify significant nodes characterizing the different T-ALL cell lines (Figure 4). Specifically, Jurkat cells showed higher phosphorylation of basal-pErk1/2, basal-pLck, basal-pAkt, basal- and H_2_O_2_-induced pZAP70 nodes. CEM cells were characterized by increased basal pJNK and pRb, and Akt induced by CXCL12 and H_2_O_2_. MOLT4 cells exhibited higher STAT3 phosphorylation in the basal condition (Figure 4). These nodes were particularly important for T-cell classification since the clustering of T-ALL cell lines was lost if they were excluded from analysis (Appendix A).

Results from independent experiments showed an overall consistency of basal and induced phosphorylation measurements among experiments. Interestingly, comparison of these data across T-ALL and HD T cells showed that Jurkat significantly contrasted HD T cells by high levels of basal-pAkt, basal-pERK1/2, basal-pZAP70^Y319^, and basal-pZAP70^Y292^ (Figure 5A). Moreover, H_2_O_2_ -induced pZAP70^Y292^ was significantly higher in Jurkat when compared with CEM (Figure 5B). Compared with normal T cells, pRb in the basal condition and CXCL12-induced pAkt were significantly higher in CEM (Figure 5). In addition, pAkt induced by H_2_O_2_ was significantly higher in CEM when compared with Jurkat cells (Figure 5B).

## 4. Discussion

Here, we apply phosphospecific flow cytometry to functionally characterize elements of the T-ALL signaling network. We measured signaling properties in the T-ALL cell lines CEM, MOLT4, and Jurkat in the basal condition and following cell modulation. Indeed, we and others have previously shown that signaling readouts in response to external modulators, rather than the basal levels of protein phosphorylation alone, are of clinical relevance in leukemia [40,41,47]. Besides the physiologic modulators CXCL12 and IL7, we used PMA to induce T-cell activation and H_2_O_2_ to inhibit protein tyrosine phosphatases (PTPs), thus enabling amplification of signaling events. Indeed, previous studies have shown that exposing cancer cells to potentiating inputs, such as H_2_O_2_, could allow us to discern cancer network profiles that correlate with disease outcome [41,48,49]. Our study shows that: (i) signaling profiles measured using phosphospecific flow cytometry are characterized by a high variability across the analyzed T-ALL cell lines; (ii) each cell line exhibits distinct intrinsic phosphorylation profiles and specific signaling sensitivity to external stimuli; and (iii) hierarchical analysis of signaling properties allows us to group T-ALL cell lines in two separate clusters.

Analysis of the phosphorylation status of signaling proteins in the unstimulated condition—which captures intrinsic signaling alterations—documents constitutive phosphorylation of pErk1/2, pNF-κB p65, pLck, ZAP70^Y319^, and pSTAT3 in all T-ALL cell lines, with a higher activation of Erk1/2, Lck, and ZAP70^Y319^ in Jurkat and higher levels of basal pSTAT3 in MOLT4 cells. Interestingly, in the presence of tyrosine-phosphatase inhibition as well as following PMA treatment, pErk1/2 further increases in CEM and Jurkat cells while pSTAT3 increases in all cell lines, with a higher response in CEM cells that in contrast exhibit low pSTAT3 in the basal condition. Moreover, inhibition of tyrosine phosphatase by H_2_O_2_ induces augmentation of pNF-κB p65 in MOLT4.

Erks are members of the MAPK family that play an important role in both regulation of T-cell receptor (TCR) signaling [50] and functional responses [51] in T cells. The MAPK/Erk signaling pathway is frequently aberrantly activated in T-ALL, as a result of IL7Rα, JAK1, KRAS, NRAS, BRAF mutations [52,53]. Accordingly, the constitutive activation of Erk1/2 in T-ALL cell lines can account for mutations of KRAS in CEM; NRAS in MOLT4, and BRAF in Jurkat [54]. The MAPK/Erk signaling pathway is central in T-ALL and it has been recently shown to drive steroid resistance by phosphorylation and inactivation of the pro-apoptotic BH3-only protein BIM [12]. Remarkably, MEK inhibitors abolishes the inactivating phosphorylation of BIM and resensitizes steroid-resistant T-ALL cells in preclinical models [12].

NF-κB family transcription factors are a common downstream target for inducible transcription mediated by many different cell-surface receptors, especially those receptors involved in inflammation and adaptive immunity [55]. In T cells, NF-κB can be activated by a variety of immune signals and is involved in early thymocyte development and in antigen-dependent T-cell selection [56,57]. Although mutations in NF-κB signaling genes are rare in T-ALL, consistently with our data, constitutive activation of the pathway is often observed in T-ALL [58,59]. NF-κB signaling can be activated by several pathways, including PI3K/Akt and NOTCH1 [13]. The PI3K/Akt signaling pathway is frequently constitutively activated in T-ALL, mainly because of the inactivation of phosphatase and tensin homolog (PTEN), a phosphatase acting as negative regulator of the PI3K/Akt pathway [60,61]. Moreover, activating mutations in *NOTCH1* have been identified in over 60% of T-ALL cases [62]. Consistently, *PTEN* inactivating mutations as well as *NOTCH1* mutations have been documented in all the T-ALL cell lines analyzed in this study [54].

T-ALL cell lines also exhibit the concurrent phosphorylation of Lck and ZAP70^Y319^, with higher phosphorylation levels observed in Jurkat cells. Interestingly, inhibition of tyrosine phosphatases further increases pZAP70^Y319^ in Jurkat. Lck is a central kinase in T-cell precursors for the transition of CD4/CD8 double negative to double positive thymocytes and proliferation of early T-cell precursors [63]. Moreover, mice transgenic for active Lck develop thymic tumors, thus suggesting that Lck plays an important role in the pathogenesis of T-ALL [64]. The Lck and ZAP70 tyrosine kinases are crucial for initiating TCR signaling because TCR has no intrinsic enzymatic activity. Upon TCR stimulation, Lck phosphorylates the TCR, thus leading to the recruitment, phosphorylation, and activation of ZAP70 [65]. Moreover, the binding of the Lck SH2 domain to pZap70^Y319^ may sustain Lck localization and the catalytic activities of both kinases, thereby providing positive feedback [66]. Therefore, our data suggest that the analyzed T-ALL cell lines show constitutive activation of phosphoproteins on the pre-TCR/TCR pathway, with higher activation levels detected in the more mature T-ALL cell line Jurkat. Interestingly, Jurkat cells also shows the constitutive phosphorylation of Zap70^Y292^, which is increased following treatment with H_2_O_2_. The Y292 residue of Zap70 is known to mediate inactivation of the TCR signaling [67]. Taken together, these data are consistent with previous studies evidencing the crucial role of the preTCR complex signaling in supporting T-ALL leukemogenesis [68,69].

In contrast to the phosphorylation pattern described for Erk1/2, Lck, and ZAP-70^Y319^ showing a higher intrinsic phosphorylation in Jurkat cells, we document a higher phosphorylation of STAT3 in MOLT4. The constitutive phosphorylation of STAT3 in all the T-ALL cell lines analyzed is consistent with previous studies showing high levels of pSTAT3 in T-ALL [70]. The apparent discordance with recent data from Bonaccorso et al. demonstrating no constitutive STAT3 phosphorylation in CEM and Jurkat cells [71] could be explained by the different STAT3 phospho-sites analyzed in the two studies (i.e., Y705 in the Bonaccorso’s analysis and S727 in our study). Remarkably, our data are coherent with the crucial role of persistently activated STAT3 in increasing tumor cell proliferation, survival and invasion while suppressing anti-tumor immunity [72].

Rb is constitutively phosphorylated in CEM and MOLT4, with CEM showing the highest levels of pRb. The tumor suppressor Rb is a negative regulator of cell cycle and Rb inactivation by deletion or hyperphosphorylation has been shown to induce cell-cycle progression in various cancers [73]. Hyperphosphorylation of Rb we document in T-ALL cell lines is in accordance with previous results demonstrating that Rb is hyperphosphorylated in most T-ALL primary samples [74] and supports the idea that deregulation of the cell cycle pathway in T-ALL is central in the pathogenesis of this disease.

Akt and JNK are constitutively phosphorylated in CEM and Jurkat cells, with Jurkat showing the higher intrinsic Akt phosphorylation. Moreover, in the presence of tyrosine phosphatase inhibition, pAkt levels are increased in CEM and MOLT4. The PI3K/Akt signaling pathway is frequently constitutively activated in T-ALL, mainly due to *PTEN* inactivating mutations [60,61], which is coherent with *PTEN* mutations documented in the analyzed T-ALL cell lines [54]. In normal T cells, the PI3K/Akt signaling pathway is a common effector cascade downstream of NOTCH, IL7, and pre-TCR signaling promoting survival and development of early T-cell precursors [75]. In T-ALL, the PI3K/Akt pathway is commonly activated generating anti-apoptotic and proliferative signals [60,61,76] and, accordingly, this signaling pathway has been explored as a novel therapeutic target in T-ALL [77]. PI3K inhibitors have shown some promising effects in T-ALL preclinical models [78]. Moreover, the Akt inhibitor triciribine induces cell cycle arrest and apoptosis and synergizes with vincristine in T-ALL cell line [79].

JNK, a member of MAPK superfamily, plays an oncogenic role in several cancers, especially in malignant lymphocytes [80,81,82]. Although the role of JNK activity in T-ALL remains largely unknown, we have previously shown that JNK activation is required for upregulating interleukin 8 expression induced by CXCL12 in primary T-ALL cells [18]. Moreover, inhibition of JNK activity has been shown to lead to cell cycle arrest and apoptosis [83,84]. Our data showing hyperphosphorylation of JNK in T-ALL cell lines suggests a role for JNK in sustaining leukemia.

Basal pSTAT5 and pp38 are both undetectable in the studied T-ALL cell lines, also in the presence of tyrosine-phosphatase inhibition and following treatment with PMA, CXCL12, or IL7.

All the analyzed T-ALL cell lines are unresponsive to IL7, probably due to the very low levels of CD127 expressed on the cell surface of these cells. Our data are in accordance with results from Bonaccorso et al., showing that basal STAT5 activation was low and nonresponsive to IL7 in PTEN-mutated T-ALL [71].

CXCL12 stimulation evokes a clear-cut increase of pAkt in CEM and MOLT4, and pJNK in MOLT4. Interestingly, Akt in CEM and MOLT4, and JNK in MOLT4 show low intrinsic phosphorylation levels. In contrast, Akt in Jurkat and JNK in CEM and Jurkat are highly phosphorylated in the basal condition. This phosphorylation pattern suggests that CXCL12 evokes Akt and JNK phosphorylation responses when these phosphoproteins are intrinsically inactive, whereas it is unable to further induce phosphorylation of Akt and JNK when they display high intrinsic activation status.

Hierarchical clustering analysis documents that higher intrinsic phosphorylation of Erk1/2, Lck, ZAP70, and Akt, together with ZAP70 phosphorylation in the presence of tyrosine-phosphatase inhibition, identifies Jurkat. Moreover, Jurkat cells exhibit higher phosphorylation levels of these nodes compared with normal T cells. In contrast, CEM are characterized by higher intrinsic phosphorylation of JNK and Rb and higher responsiveness of Akt to external stimuli, which are significantly higher in comparison with normal T cells. MOLT4 cells are characterized by higher STAT3 phosphorylation in the basal condition.

Our study originally shows that phosphospecific flow cytometry allows us to capture signaling network variability across different T-ALL cell lines. Although the use of cell lines does not allow us to relate signaling properties with disease characteristics, the ability to detect signaling heterogeneity among T-ALL cell lines—which can reflect biological heterogeneity of primary cells from which they derive—provides an important “proof of concept” for future studies aimed at characterizing signaling heterogeneity across primary leukemic cell samples. Moreover, these results corroborate genetic landscape data, suggesting that distinct signaling pathways are activated in subtypes of T-ALL [85].

Overall, the T-ALL cell populations we characterize in this study exhibit homogeneous phenotypes and monomodal distributions of the analyzed phosphoproteins. Therefore, herein we could not characterize heterogeneity at the single cell level within cell-line populations. However, phospho-specific flow cytometry is an especially useful tool for interrogating the physiology of signaling pathways by measuring network properties at the single-cell level, thus understanding the biology of heterogeneous populations of cells, such as those found in tumor samples from patients. Indeed, the signaling profile of any given leukemic cell is the sum of numerous influences—epigenetic, genetic and microenvironmental—and the heterogeneity of cancer cell behavior can be thought of reflecting the signaling differences that have arisen during the evolution of leukemic cell population. Therefore, integration of genomic and epigenetic studies with functional proteomic approaches at the single-cell level is fundamental to identify potential drug targets or biomarkers.

The functional flow cytometry approach we describe in this study provides a powerful tool for characterizing T-ALL heterogeneity across samples and has the potentiality to characterize signaling heterogeneity at the single level within primary leukemic cell populations. Therefore, it could represent an important tool for future studies on T-ALL primary cells aimed at identifying signaling elements as potential precision drug targets or biomarkers.

## 5. Conclusions

In this study, we demonstrated that phospho-specific flow cytometry can identify intrinsic and extrinsic signaling pathway activities and capture signaling network variability across different T-ALL cell lines. Overall, our study can contribute to a deeper understanding of the complex signaling network altered in T-ALL. Moreover, our data can form the basis for future studies on primary T-ALL cells connecting signaling properties with clinical behaviors and aimed at developing precision medicine tools, particularly for resistant and high-risk T-ALL patients.

## Figures and Tables

**Figure 1 cells-11-02072-f001:**
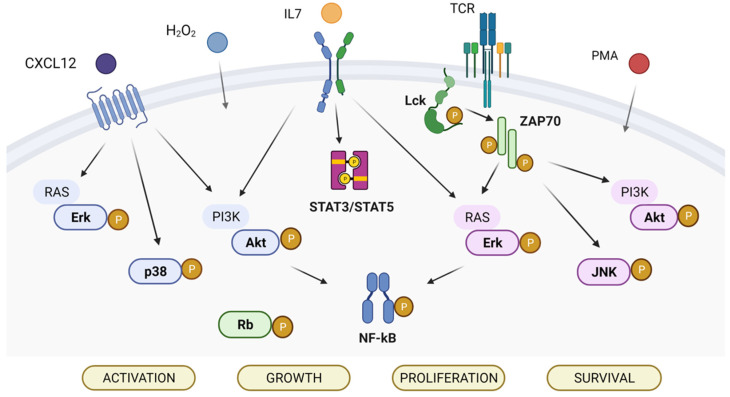
Signaling pathways analyzed in the study. The analyzed phosphoproteins are highlighted in bold.

**Figure 2 cells-11-02072-f002:**
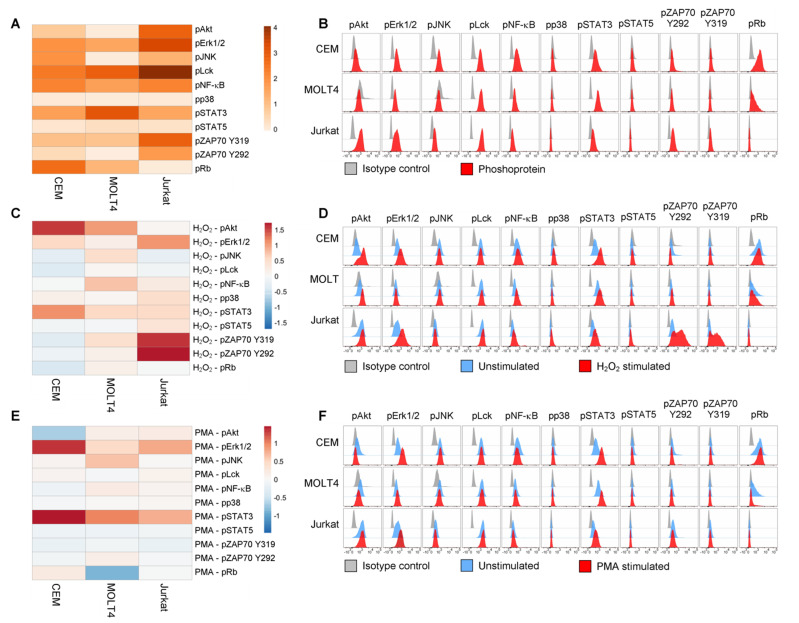
Phosphorylation profiles of T-ALL cell lines. (**A**,**B**) Pseudocolor heatmap (**A**) and flow cytometry histograms (**B**) of all the analyzed phospho-proteins in basal condition (unstimulated) in T-ALL cell lines. (**C**,**D**) Pseudocolor heatmap (**C**) and flow cytometry histograms (**D**) of the phosphorylation profiles following modulation with 3.3 mM H_2_O_2_ for 15 min. (**E**,**F**) Pseudocolor heatmap (**E**) and flow cytometry histograms (**F**) of the phosphorylation profiles following modulation with 400 nM PMA for 15 min. Data are representative of three independent experiments.

**Figure 3 cells-11-02072-f003:**
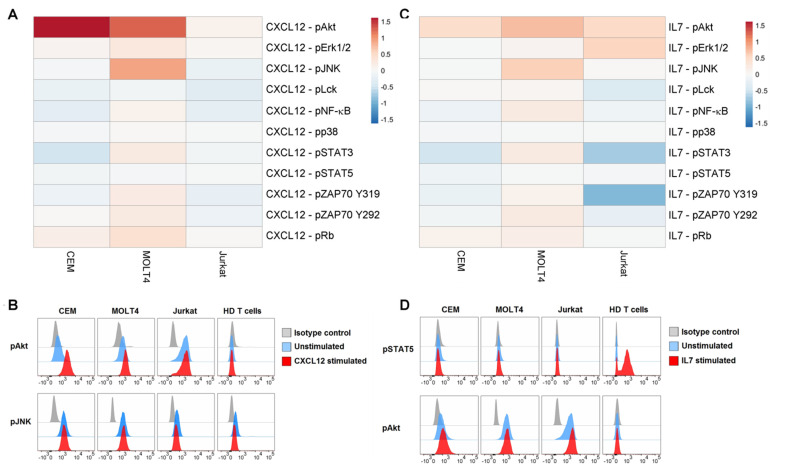
Phosphorylation profiles following modulation with CXCL12 or IL7. (**A**) Pseudocolor heatmap of the phosphorylation profiles following modulation with 100 ng/mL CXCL12 for 15 min. (**B**) Flow cytometry histograms of pAkt and pJNK in basal condition (unstimulated) or after CXCL12 modulation in T-ALL cell lines and T cells from HDs, compared with isotype signals. (**C**) Pseudocolor heatmap of the phosphorylation profiles following modulation with 10 ng/mL IL7 for 15 min. (**D**) Flow cytometry histograms of pSTAT5 and pAkt in basal condition (unstimulated) or after IL7 modulation in T-ALL cell lines and T cells from HDs, compared with isotype signals. Data are representative of three independent experiments.

**Figure 4 cells-11-02072-f004:**
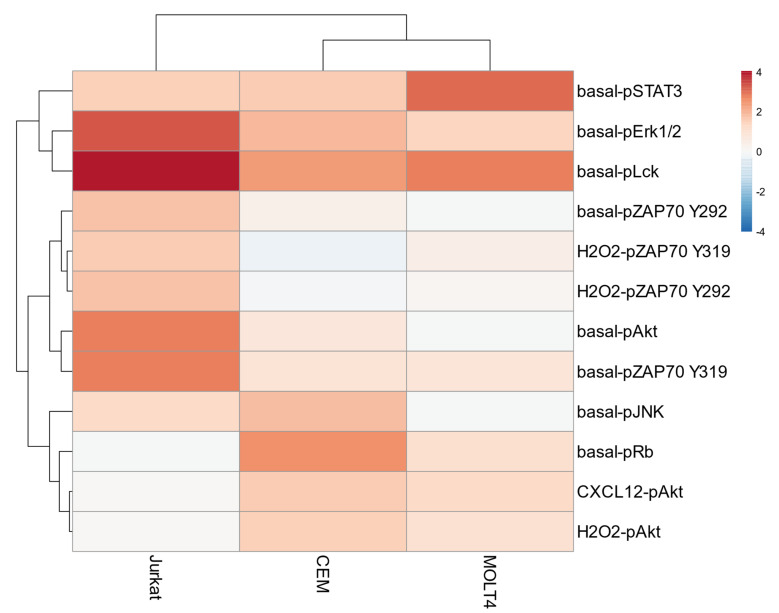
Unsupervised hierarchical clustering analysis on selected nodes. Unsupervised hierarchical clustering analysis of nodes showing higher variation (σ^2^ of log_2_ values > 0.5; *n* = 12) across samples within the T-ALL cell lines CEM, MOLT4, and Jurkat. Data are representative of three independent experiments.

**Figure 5 cells-11-02072-f005:**
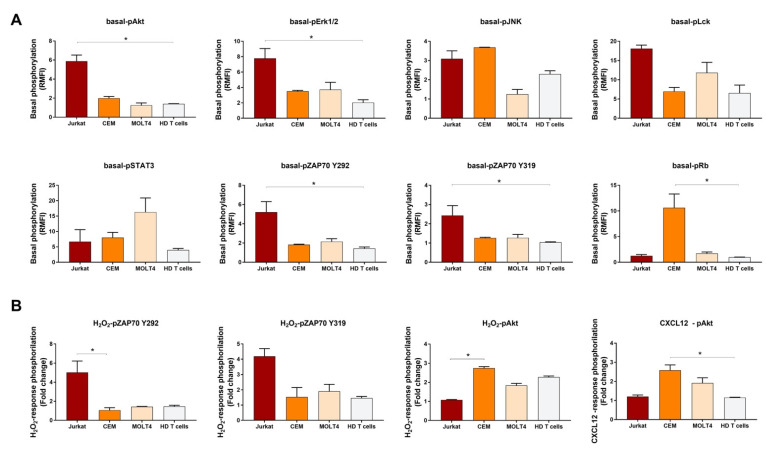
Histograms of phosphorylation levels in T-ALL cell lines and HD T cells. (**A**) Comparison of phosphorylation levels in the basal condition, relative to isotypic controls, among T-ALL cell lines and healthy donor (HD) T cells. (**B**) Comparison of response to stimulation, relative to the unstimulated condition, among T-ALL cell lines and HD T cells. Data are expressed as mean ± SEM from three, independent experiments. Comparisons were performed using the Kruskal–Wallis test and each *p* value was corrected for multiple comparisons using the Dunn’s test (* *p* ≤ 0.05).

**Table 1 cells-11-02072-t001:** T-ALL cell lines according to the EGIL classification.

T-ALL Cell Lines	EGIL Classification	Markers		
		CD7+/cCD3+		
		CD2 and/or CD5 and/or CD8	CD1a	mCD3
	Pro-T (T I)	Neg	Neg	Neg
CEM	Pre-T (T II)	Pos	Neg	Neg
MOLT4	Cortical T (T III)	Pos	Pos	Neg
Jurkat	Mature T (T IV)	Pos	Neg	Pos

cCD3: cytoplasmic CD3; mCD3: membrane CD3; Neg: negative; Pos: positive.

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
