# Peer review of "Phospho-Specific Flow Cytometry Reveals Signaling Heterogeneity in T-Cell Acute Lymphoblastic Leukemia Cell Lines"

_cells, 2022, doi:10.3390/cells11132072_

Round 1

Reviewer 1 Report

I have already received this manuscript for revision before, and authors have answered all my concerns in this new version of the manuscript.

Author Response

No response is required.

Reviewer 2 Report

I was happy to see that the authors have responded to most of my concerns. They have confirmed that their results are the product of repeated independent experiments, they have avoided the use of the term “T-ALL cells”, they have added more data regarding the laboratory methods used, they have considerably improved their figure and have enriched their conclusion section.

Although their work has not actually reached any remarkable conclusion, the volume of generated information is impressive and may form a basis for further studies. Publishing these results will enable future researchers to avoid repetition of similar experiments and alternatively apply modified or enriched protocols in their quest for specific T ALL signaling patterns.

That is why I believe that the strength of this manuscript is based on the detailed description of laboratory methods and on the soundness and quality assurance of the results. In this context I have the following concerns:

1.      The authors have responded that their results are the product of repeated independent experiments. They have added this information in the figure legends. However, I believe that his information should be also stated in the method section.

2.      Moreover, and more importantly, calculating the mean result of 3 experiments, as the authors have done, is not optimal. What was the variation of the results? Were there any discrepancies between measurements? (ie one increased and 2 diminished responses to the same stimulus?). These data are indicative of the quality of the experiments and the soundness of the results. However, they may be masked by the mere use of mean MFI and there would always be the fear that the reported heterogeneity is nothing more that random results of poorly standardized methods. Given the overall quality of the paper and the authors’ publication history, I believe that this is not the case.  Still, I propose that it must be proven. I would therefore suggest that the authors (1) report if there were any significant discrepancies in between independent experiments and discard the relevant results (or give a reason why they would be accepted) or specifically state in the main text that there were no such discrepancies and (2) give the range of the measurements (ie min/max as n is too low for reporting SD…) in the supplement section.

3.      3. A minor comment: I still believe that in line 291 it should be shown that 2 /3 studies (ref 41 ,42) were from the authors’ center. It may seem trivial, but from my point of view, this kind of inaccuracies have a negative influence on the overall credibility of a paper.

4.      4. Although I am not a native English speaker, I believe that the use of English needs to be improved, especially in the recently added phrases.

Author Response

  1. The authors have responded that their results are the product of repeated independent experiments. They have added this information in the figure legends. However, I believe that this information should be also stated in the method section.

Response. This information has been also stated in the method section.

  1. Moreover, and more importantly, calculating the mean result of 3 experiments, as the authors have done, is not optimal. What was the variation of the results? Were there any discrepancies between measurements? (ie one increased and 2 diminished responses to the same stimulus?). These data are indicative of the quality of the experiments and the soundness of the results. However, they may be masked by the mere use of mean MFI and there would always be the fear that the reported heterogeneity is nothing more that random results of poorly standardized methods. Given the overall quality of the paper and the authors’ publication history, I believe that this is not the case. Still, I propose that it must be proven. I would therefore suggest that the authors (1) report if there were any significant discrepancies in between independent experiments and discard the relevant results (or give a reason why they would be accepted) or specifically state in the main text that there were no such discrepancies and (2) give the range of the measurements (ie min/max as n is too low for reporting SD…) in the supplement section.

Response. We agree with the Reviewer on the importance of standardization and quality control procedures in flow cytometry and more specifically in phospho-specific flow cytometry, which often measures small variation in fluorescence intensity. Therefore, as standard procedure for  phospho-specific flow cytometry analysis we monitor intra- and inter-cytometer variance and longitudinal consistency of instrument performance using a single lot of 8-peak RCP beads for each experiment. Moreover, we monitor overall assay performance by running on each experiment cell lines from a single batch that were expanded in culture, cryopreserved, and quality control tested. 

Flow cytometry data are derived from fluorescence measures carried out on a sample of 10,000 cells that, from the statistical point of view, can be considered large enough to approximate the true fluorescence distribution of the population from which that sample has been drawn.  Nonetheless, we repeated the measures in three independent experiments carried out in different dates and with different cell samples to take into account biological variability across cell generations (cell lines) or across donors (healthy cells) and the mean values of each fluorescence distributions have been averaged. According to the central limit theorem, this average tends to the true mean of the population from which all samples have been drawn, in spite of biological heterogeneity and with an uncertainty that is formally measured by the standard error of the mean (SEM). While Figures 1-4 report data from a representative experiment of three, in Figure 5 we reported the summary of independent experiments analyzing those signaling nodes that are important for characterizing heterogeneity across cell lines, as defined in Figure 4. Specifically, for each node Figure 5 reports the mean (n = 3) of the relative mean fluorescence intensity (RMFI) or the Fold change of RMFI of the whole cell populations (for each cell population, n = 10,000), respectively for the basal condition or for the modulated condition. The variation of these data is calculated as SEM. Therefore, information of data variation requested by the Reviewer is already provided in Figure 5. Moreover, overall data across the three experiments are coherent, i.e., the three measures go in the same direction, as shown by the mean values and relatively small SEM of Figure 5,. We stated that in the manuscript.

  1. A minor comment: I still believe that in line 291 it should be shown that 2/3 studies (ref 41 ,42) were from the authors’ center. It may seem trivial, but from my point of view, this kind of inaccuracies have a negative influence on the overall credibility of a paper.

Response. We added a sentence that provides this information.

  1. Although I am not a native English speaker, I believe that the use of English needs to be

Response. The manuscript has been checked by a native English-speaking lecturer from our university.  

Reviewer 3 Report

The authors have addressed the questions.

Author Response

No response is required. 

Thank you.

Round 2

Reviewer 2 Report

I thank the authors for their detailed response regarding quality assurance in their FC experiments.

Although I find that the presentation of the results is of high quality, I believe that laboratory performance quality issues were hidden behind the bulk of information provided in this manuscript. I regret that I myself did not notice the SEM on figure 5, despite thouroughly and repeatedly reading the manuscript. I therefore feel that it was helpful to indicate this in the main text.

Overall I believe that this is a very interesting and detailed work that will help further studies in the field.